

# The Variable Infiltration Capacity Model, Version 5 (VIC-5): Infrastructure improvements for new applications and reproducibility

Joseph J. Hamman[1,2], Bart Nijssen[1], Theodore J. Bohn[3], Diana R. Gergel[1], and Yixin Mao[1]

[1]Department of Civil & Environmental Engineering, University of Washington, Seattle, WA, USA.
[2]Now at: Hydrometeorological Applications Program, Research Applications Laboratory, National Center for Atmospheric Research, Boulder, Colorado, USA
[3]School of Earth and Space Exploration, Arizona State University, Tempe, AZ, USA.

*Correspondence to:* Bart Nijssen (nijssen@uw.edu)

**Abstract.** The Variable Infiltration Capacity (VIC) model is a macro-scale semi-distributed hydrologic model. VIC development began in the early 1990s and the model has since been used extensively for basin- to global-scale applications that include hydrologic data set construction, trend analysis of hydrologic fluxes and states, data evaluation and assimilation, forecasting, coupled climate modeling, and climate change impact assessment. Ongoing operational applications of the VIC model include
the University of Washington's drought monitoring and forecasting systems and NASA's Land Data Assimilation System. This paper documents the development of VIC version 5 (VIC-5), which includes a major reconfiguration of the legacy VIC source code to support a wider range of modern hydrologic modeling applications. The VIC source code has been moved to a public GitHub repository to encourage participation by the broader user and developer communities. The reconfiguration has separated the core physics of the model from the driver source code, where the latter is responsible for memory allocation, pre- and post-processing and input/output (I/O). VIC-5 includes four drivers that use the same core physics modules, but which
allow for different methods for accessing this core to enable different model applications. Finally, VIC-5 is distributed with robust test infrastructure, components of which routinely run during development using cloud-hosted continuous integration. The work described here provides an example to the model development community for extending the life of a legacy model that is being used extensively. The development and release of VIC-5 represents a significant step forward for the VIC user
community in terms of support for existing and new model applications, reproducibility, and scientific robustness.

## 1 Introduction

The Variable Infiltration Capacity (VIC) model (Liang et al., 1994) is a semi-distributed, macro-scale hydrologic model (MHM) that has been applied in a broad set of use cases. The model has been used for numerous water and energy balance studies in the U.S. (Abdulla and Lettenmaier, 1997; Nijssen et al., 1997), the Arctic (Adam and Lettenmaier, 2008; Su et al., 2005;
Tan et al., 2011; Hamman et al., 2016) and globally (Nijssen et al., 2001a, b, c; Sheffield et al., 2009). One result of these studies has been refinement of the model to better represent key hydrological processes (Andreadis et al., 2009; Cherkauer et al., 2003; Liang et al., 1996, 1999). Ongoing realtime applications of the VIC model include the University of Washington's





drought monitoring and forecasting systems (http://hydro.washington.edu/forecast/monitor/drought/index.shtml), and NASA's Land Data Assimilation System (LDAS; https://ldas.gsfc.nasa.gov). Table 1 provides additional examples of the use of VIC for hydrological, water resources, and water and energy budget applications.

Although the motivation for the development of VIC, as documented in the original VIC publication (Liang et al., 1994),
was as a land surface scheme for coupled land-atmosphere models and earth system models (ESMs), the VIC model has been applied predominantly in uncoupled modeling studies in which there is no feedback from the land surface to the atmosphere. VIC and other uncoupled MHMs have continued to exist alongside the fully-coupled land surface schemes and have developed a large user community for a number of reasons: a) their focus on the representation of hydrological processes often resulted in simulations that were hydrologically more credible than the land surface schemes typically used in coupled land-atmosphere
models (as shown, for instance, in the Project for Intercomparison of Land Surface Parameterizations (PILPS; Bowling et al., 2003; Wood et al., 1998)); b) their ability to simulate variables that are directly relevant for water resources management, such as streamflow, makes them useful for planning and impact studies; and c) their lower demand for computational resources (compared with coupled models) allows research groups that do not have access to supercomputers to perform large-scale simulations. Most of these uncoupled MHMs can, for most applications, be run on single processors or small computer clusters.
Because VIC was traditionally operated as a stand-alone land surface scheme, there were no requirements for the model code to communicate with other model components in a coupled environment. In other words, there was no requirement for the model to *"play well with others"*. Instead, the model run-time environment focused on simplicity with the goal that VIC could run without the need for specific machine architectures, compilers or third-party libraries. Model development prior to version 5 did not follow a well-defined development track and was directed primarily by the needs of specific model experiments.
Relatively little attention was paid to improvements of the model infrastructure, which had remained essentially unchanged since its initial release in the early 1990s, including input/output (I/O) formats. As a consequence, model implementation typically required extensive pre- and post-processing, including reformatting and processing of model input and output files. The few VIC applications within multi-model frameworks (e.g. NASA LIS, Kumar et al., 2006) required significant custom reconfiguration of the model interface. While VIC model source code was version controlled and publicly available since the
early 1990s, model development was not always transparent. The source code repository of record resided on a private machine at the University of Washington, where most of the development was conducted.

Although the ability to run MHMs such as VIC in uncoupled mode has been one of their strengths, earth system science has reached a point where the inability of some MHMs to be used in both coupled and uncoupled mode is a serious shortcoming. While a few previous efforts have been made to couple VIC to both atmospheric and fully-coupled earth system models
(e.g. Zhu et al., 2009; Hamman et al., 2016), these methods have been ad-hoc and required significant rewriting of the model infrastructure. In effect, this resulted in a version of the model code that was coupled, but which would rapidly become obsolete relative to the uncoupled VIC version which was the main path for model enhancement and improvement.

Interest in and advocacy for the use of established software development methods has grown in recent years in the fields of hydrology and the earth sciences to promote reproducibility of model and analysis results (e.g. Wilson et al., 2014; Ceola et al.,
2015; Fienen and Bakker, 2016; Gil et al., 2016; Hutton et al., 2016). Opinions differ with respect to the best workflows to





promote reproducibility. For example, an active discussion ensued in response to the paper by Hutton et al. (2016), not so much about the need for improved workflows, but about implementation details and the required level of rigor (Anel, 2017; Melsen et al., 2017; Hut et al., 2017; Hutton et al., 2017a, b). This debate takes place within a broader academic discussion of the need for open-source computer programs to increase scientific transparency in the model development and application process
(Ince et al., 2012). Improved model source code maintenance, standardized test setups, and transparency comprise only part of this drive towards greater reproducibility. Other aspects include model and workflow documentation, versioning of data sets, and inclusion and standardization of metadata. It is with these points in mind that many of the new process and infrastructure improvements discussed in this paper were developed.

This paper documents the development of VIC version 5, hereafter referred to as VIC-5, a new major release of the VIC
model. The motivation for the development of VIC-5 is to extend VIC's useful life by adding the necessary infrastructure to allow the model to operate in a number of coupled and uncoupled modes, while relying on the same underlying physics implementations, and to ensure that upgrades to the physics routines are automatically available to all model configurations. In this paper we describe the software, infrastructure, and associated improvements in VIC-5 and provide baseline model diagnostics in terms of computational performance. The initial VIC-5 release does not add new or improved physical process
representations compared to the last release of the VIC-4 development track (version 4.2.d). Instead, the development of VIC-5 focused on reconfiguring the legacy VIC source code to support a wider range of modeling applications, to better integrate VIC with other applications, and to enhance reproducibility and source code maintenance. Therefore, this paper does not attempt to provide a scientific benchmarking of the VIC model. We have, however, provided a summary table of the key improvements to the VIC model since the original Liang et al. (1994) paper in Appendix A.

This paper is organized as follows. Sect. 2 briefly outlines the VIC legacy infrastructure in the model versions before VIC-5, details our design objectives, and discusses the major infrastructure changes implemented to meet those objectives. Sect. 3 includes results from test simulations with the new model code. The results and discussion from these test simulations are presented in Sect. 4. Concluding remarks are provided in Sect. 5 along with a discussion of our efforts to create and cultivate a VIC user community and to share the model source code.

## 2   VIC Infrastructure

### 2.1   Legacy Implementation

Although VIC was originally intended to function as a land surface component in ESMs (Liang et al., 1994), its early development as a stand-alone model led to a model infrastructure that was poorly suited to coupling with atmospheric models. Most importantly, because model physics did not account for fluxes between adjacent grid cells (as was and remains common among
land surface components of ESMs), VIC used a *time-before-space* computation order in which all time steps for a given model grid cell were completed before the model moved to the next location. We will refer to the *time-before-space* configuration as *vector* mode. This scheme is in direct conflict with atmospheric models, which use a *space-before-time* computation order in which all grid cells in the entire model domain are simulated for a single time step before advancing to the next time step. We



will refer to the *space-before-time* configuration as *image* mode. In addition, consistent with the vector mode implementation, VIC's inputs and outputs consisted of one or more (ASCII or binary) files per grid cell, each containing the entire timeseries of a cell's fluxes and states. While this kept VIC's memory requirements relatively low, it led to an unwieldy number of files as domain size and grid resolution increased. Furthermore, analyzing, plotting, and publishing results from these files often

5 required a time-consuming conversion to more flexible formats such as Unidata's Network Commmon Data Format (NetCDF; Rew and Davis, 1990). The challenge presented by separate files for each model grid cell is insidious. While it makes it easy to examine time series for individual grid cells, it makes it more difficult to examine time series of domain-wide model output and can hide errors in spatial model inputs and outputs that may be readily apparent upon inspection of a time series of spatial output.

10 While VIC's original infrastructure may have made coupling within ESMs difficult, it also contributed to the model's wide uptake throughout the hydrologic modeling community. Previous versions of VIC were very portable, required no third-party libraries, were easy to run in single point mode for testing or comparison with point observations, and were trivial to parallelize across many processors and machines by manually subdividing the domain and running each sub-domain on a separate processor. It was this simple infrastructure, combined with the inclusion of built-in convenience tools such as the meteorological

15 variable disaggregator based on the MT-CLIM algorithms (Thornton and Running, 1999; Bohn et al., 2013), publicly available regional and global forcing and parameter datasets, and publicly available source code, that greatly contributed to VIC's wide adoption in the hydrologic modeling community.

## 2.2 Design goals for VIC-5

The VIC-5 release includes a complete refactor of the model source code. The design goals of the refactor included:

- separation of the model physical core from the driver code;

- multiple model drivers to support a range of modeling applications;

- parallel processing to facilitate efficient large domain simulations;

- machine-independent and self-documenting I/O formats such as NetCDF;

- model extensions framework to facilitate coupling of sub-model components;

- comprehensive test suite to guide future model use and development;

- source code availability and version control via an online platform to enable community participation in model maintenance and development, and to promote transparency;

- improved model documentation.

The implementation details for each of these topics are discussed in the following sections.

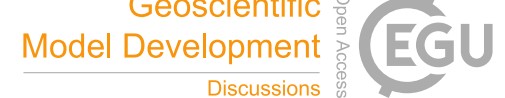

A practical motivating factor in the development of VIC-5 was to enable coupling of VIC within the Regional Arctic System Model (RASM; Hamman et al., 2016), a regional implementation of the Community Earth System Model (CESM; Hurrell et al., 2013), while maintaining usability as a stand-alone model. In RASM, VIC-5 is directly coupled to CESM's flux coupler (CPL7; Craig et al., 2012) via the Model Coupling Toolkit interface (MCT; Larson et al., 2005). As implemented within RASM, this interface allows VIC to be coupled to the Weather Research and Forecasting (WRF) atmospheric model (Skamarock, 2008) and the RVIC streamflow routing model (Hamman et al., 2017) components, while maintaining the legacy and stand-alone model infrastructure, which have a large existing user base. The successful coupling of VIC-5 within RASM is a key demonstration of the benefits of the source code refactor. The new infrastructure will facilitate VIC to be coupled more readily to other modeling systems in the future.

## 2.3 Drivers

The most significant change in the model reconfiguration was the separation of the physical model core from the model drivers, which are responsible for memory allocation, pre- and post-processing, and input and output (I/O). The VIC-5 source code structure is depicted in Fig. 1. The initial VIC-5 release includes four drivers, each using the same physical model core:

1. The *Classic Driver* supports legacy VIC configurations and runs in the traditional time-before-space or vector mode. It uses ASCII and flat-binary file formats for I/O. With the exceptions of the removal of the forcing disaggregator and minor changes to formatting and naming of input options, the *Classic Driver* is backwards-compatible with VIC version 4.2.d. More discussion on the removal of the forcing disaggregator from the *Classic Driver* is provided in Sect. 2.6.

2. The *Image Driver* includes a stand-alone space-before-time configuration, NetCDF I/O, and uses the message passing interface (MPI; Gropp et al., 1996) and OpenMP (Dagum and Menon, 1998) standards for parallel processing. The parallel processing functionality combined with NetCDF I/O has significantly streamlined VIC simulations over large spatial domains. This configuration also facilitates the direct coupling within VIC of streamflow routing, reservoir, and irrigation processes, all of which require spatial knowledge of hydrologic conditions as the model advances in time.

3. The *CESM Driver* couples VIC to the Community Earth System Model's (CESM; Hurrell et al., 2013) flux coupler (CPL7; Craig et al., 2012) and a prognostic atmosphere. The current application of the *CESM Driver* is within the Regional Arctic System Model (RASM; Hamman et al., 2016), a regional implementation of CESM. Future applications of the *CESM Driver* could be applied within the global version of CESM. Furthermore, the CESM driver may serve as a template for other coupled model applications as it provides the infrastructure necessary to couple VIC, which is written in C, to libraries and model components written in Fortran via a standardized interface. The *CESM Driver* shares much of its foundational infrastructure with the *Image Driver*. In particular, they share common source code for their parallelization and I/O layers. See Fig. 1 for illustration of the shared infrastructure between the Image Driver and the CESM Driver.



4. The *Python Driver* provides Python bindings to the functions and data structures of VIC's physical core. These functions are made available as Python functions using the C Foreign Function Interface for Python (CFFI; http://cffi.readthedocs. io). Although the *Python Driver* is currently only used for testing, this driver is uniquely suited for future interactive classroom sessions, data assimilation, or large ensemble applications.

## 2.4 Extensions

In VIC-5, we have introduced a model extensions framework to facilitate the coupling of sub-model components. Previous versions of VIC have added extended functionality to the model such as streamflow routing (Lohmann et al., 1996; Hamman et al., 2017), reservoirs, irrigation (Haddeland et al., 2006), glaciers and ice sheets, and dynamic vegetation and crops (Adam et al., 2015). These extensions were added either as stand-alone post-processing options or via highly customized interfaces. The motivating factor for developing a model extensions framework was to allow for the optional coupling of existing sub-models that are beyond the scope of core VIC applications or that are not feasible to add for all drivers.

The updated model configuration facilitates the direct coupling of new and existing subcomponents to VIC. Many of the previously developed sub-model components required information from neighboring grid cells or upstream processes (e.g. reservoir operations, irrigation, etc.). Leveraging the new functionality within the *Image Driver*, VIC is now able to provide information of storages and states from anywhere in the model domain to these sub-models. As of VIC version 5.1, only the streamflow routing extension, which couples the RVIC streamflow routing model (Hamman et al., 2017) to the *Image Driver*, is complete.

A key concept in the design of the VIC extensions framework is to allow for modular use and development of specific extensions. Therefore, each extension is enabled via compile-time options with the requirement that the model be able to run with or without the extension turned on. When turned off, each extension provides stub, or no operation, functions that allow the model to run without impacting the simulation. When turned on, each extension may use or modify state and flux variables from the core VIC model.

## 2.5 Parallel Computing

The VIC model does not include horizontal exchange between grid cells. In the past, this feature has allowed users to manually parallelize their VIC applications by running VIC simulations for different grid cells on separate cores or computers (a technique referred to as *poor man's parallelization*). While this approach generally worked well for increasing the throughput of model simulations, it required laborious and time-intensive post-processing of the model output. Often times, the time required to post-process these VIC simulations exceeded the model run time. In the image and CESM drivers in VIC-5, we have added formal parallel processing support by utilizing the MPI standard and shared memory threading via OpenMP. The parallel scaling performance of the VIC-5 image driver is discussed in detail in Sect. 3.

Because VIC model grid cells do not exchange water or energy with their neighbors, each grid cell can be modeled independently. Our MPI parallelization strategy is to use MPI only to facilitate I/O operations. Thus, all I/O is performed by the master process. Inputs are read and scattered to the individual processors, while outputs are gathered and written out by the master




process. The default domain composition uses a simple round-robin approach in which each grid cell in the model domain is sequentially distributed to another processor. For example, in the case of four MPI processes and 20 model grid cells, the first MPI process would handle grid cells 1, 5, 9, 13, and 17; the second would handle 2, 6, 10, 14, 18; and so forth. In practice, this ensures a similar computational load for each MPI process for the most common VIC spatial configuration (a regular 2-D grid).

We have also implemented an alternative parallelization strategy using OpenMP threading that may be used for concurrent simulation of multiple grid cells within a node or process. As opposed to the full MPI parallelization, OpenMP avoids some of the overhead inherent in MPI (e.g. communication, data transfer). On some machines, the use of hyper-threading with OpenMP may result in better scaling than with MPI. Finally, the addition of OpenMP allows for *Hybrid OpenMP-MPI* parallelism wherein OpenMP and MPI are used in concert. This advanced form of parallelism utilizes MPI for inter-node communication and OpenMP for shared-memory threading on-node.

### 2.6 Input and Output

The *Image* and *CESM drivers* use Unidata's NetCDF library for all I/O related to input parameters, meteorological forcings, and model output. NetCDF is a software library that reads and writes self-describing array-oriented scientific data with its metadata. It has been widely adopted among the geo-scientific modeling community and provides a convenient self-describing machine-independent data format. Interfaces to the NetCDF library are available in many scripting languages including Python, R, Matlab, and Julia. Output files from the *Image* and *CESM drivers* are formatted such that they meet the NetCDF Climate and Forecast (CF) metadata conventions version 1.6 (Eaton et al., 2003).

All drivers share a common logging module that greatly improves VIC's ability to self-document each model experiment. This logging module records the model version, input files, model settings, and performance statistics from key parts of the model source code. The verbosity of the logging module is controlled by a compile time option. During model development, the logging module also facilitates debugging by providing full stack traces and line numbers when the model exits unexpectedly. When using MPI for parallelization, the logging module provides MPI task granularity when logging messages.

The *Classic Driver* continues to support the legacy ASCII and raw binary input and output file formats. We have also added additional metadata to the header of the ASCII file format but it has basically remained the same since VIC-4.

VIC previously included a meteorological forcing processor that made use of the MT-CLIM algorithms to estimate radiation and humidity from daily inputs of precipitation and temperature and which created sub-daily forcings from daily inputs. In VIC-5, this forcing processor has been removed from the model to facilitate a number of key enhancements that are described below:

1. *Exact restarts*: In the versions of VIC that included MT-CLIM, exact restarts were not possible due to a rolling time average in MT-CLIM. In practice, applications that required exact restarts for short model runs (e.g. hydrologic forecasting) were forced to run the model for long spin-up periods to achieve near-exact restarts.



2. *Identical treatment of forcings between drivers*: The development of the *Image Driver*, which runs in a space-before-time loop order, was not compatible with MT-CLIM's loop order, which requires a time-before-space order.

3. *Improved transparency in the forcing generation process*: While VIC's core function is as a hydrologic model, specific modeling decisions were made when MT-CLIM was chosen as VIC's meteorological preprocessor and those decisions were abstracted deep within the model source code. As a result, the implications of forcing VIC with only minimum and maximum temperature and precipitation were abstracted away from and mostly invisible to the user. Removing the preprocessor from VIC enhances the transparency of the model by forcing users to make specific choices about the source of the model forcings.

A result of this change is that the input forcings must now have the same time step length as the model simulation. Consequently, for offline simulations based on widely-used gridded daily observations (e.g. Livneh et al., 2015), forcing disaggregation must occur outside the model as a pre-processing step. A Python-based stand-alone disaggregation tool, MetSim (http://metsim.readthedocs.io), has been developed for this purpose.

VIC-5 also has eliminated all hard-coded model parameters (e.g. emissivity of snow and the temperature lapse rate) from the body of the source code and collected physical constants into a single header file. Users may now specify an optional "parameters" configuration file to override default model constants that were previously hard-coded values. These changes follow a push in the hydrologic modeling community for hydrologic models to expose constant parameters (e.g. Mendoza et al., 2015), and greatly improves the transparency, flexibility, and accessibility of the model.

## 2.7 Testing

VIC-5 introduces a comprehensive test suite, now distributed with the model source code. The VIC test suite was designed to serve three primary purposes: 1) automated diagnostics and benchmarking of model performance, following Luo et al. (2012); 2) continuous integration providing automated build and unit tests to ensure reproducibility; and 3) enable development contributions from a broader user community by facilitating consistency between model developers. As the VIC developer community continues to grow, this test suite will be an important tool in maintaining the function of the supported drivers.

The test suite is made up of seven components:

1. *Build*: The build tests ensure that VIC may be compiled and linked using a variety of different libraries and compilers. Given the increased complexity of the software stack supporting VIC, these tests are important to ensure portability of the model.

2. *Unit*: The unit tests check function-level source code behavior. Using the *Python Driver*, many of the individual functions in VIC are tested using the "Py-Test" unit test framework (http://doc.pytest.org).

3. *System*: The system tests check the behavior of the individual drivers. Examples of these tests include checks for exact restarts using model-generated state files, binary equivalence for simulations run with and without parallel processing, and comparisons between drivers.



4. *Science*: The science tests utilize the *Classic Driver* to run point simulations at a selection of observation locations (e.g. SNOTEL or AMERIFLUX sites). Analysis figures and summary statistics are made as part of the automated test framework. Results from these point simulations are quantitatively and qualitatively compared to observations and also to simulation results from previous VIC versions. Figs. 2 and 3 provide examples of aggregated output from the science test suite. Additional discussion related to these figures is provided in Sect. 4.

5. *Examples*: A set of short example VIC configurations and inputs (parameters and forcings) are provided to users for the *Classic* and *Image drivers*. The *Examples* tests check that the example data continue to run such that new users can quickly and easily get started with the VIC model.

6. *Release*: The release tests are longer, full domain (e.g. North America, global, etc.), *Image Driver* simulations performed prior to each model release. These tests demonstrate the current model behavior and are compared to previous releases to evaluate the changes in the model physics. These tests are archived for each model release to allow for retrospective model comparison.

7. *Performance*: With the addition of parallel processing to the *Image* and *CESM drivers*, a set of tools for assessing the parallel performance were needed. The performance tests assess timing and memory statistics of the VIC model. Results from these tests are highlighted in Sect. 4.

The *Build*, *Unit*, *System*, and *Examples* tests are routinely run using the Travis CI continuous integration system (https://travis-ci.org/UW-Hydro/VIC) where the VIC test suite is executed using a variety of compilers and computational architectures. The VIC's Travis workflow includes the following steps: 1) selection of virtual machine with preinstalled environments, 2) download of VIC source code and test datasets from GitHub, 3) compile specific driver along with Python unit tests, and 4) run series of tests for specific driver implementation (e.g. image). If any of the tests in these steps fail for any part of the test matrix, Travis returns a failed test status to the VIC GitHub page. This functionality is routinely used in evaluating VIC modifications that are suggested on GitHub via pull requests.

## 2.8 Model Distribution and Documentation

Beginning in 2013, the VIC source code has been publicly available on GitHub (https://github.com/UW-Hydro/VIC). GitHub is a web-hosting service built on top of the Git version control system and is meant to facilitate social coding. VIC development uses many of the features available from GitHub including user and administrator controls, code review tools, and a bug and issue tracker. The ability to provide detailed, line-by-line code reviews, has been particularly useful in the process of developing VIC-5 with multiple developers who have varying familiarity with the VIC source code.

We have adopted the widely used "Gitflow" branching model and workflow for managing the VIC Git repository. The workflow provides guidance for managing development branches and releases. Key features of the workflow are the separation of the master branch from development, feature, support, and bug-fix branches. This workflow is described in detail as part of VIC's online documentation.





Finally, VIC has adopted the semantic versioning system (Preston-Werner, 2018) for determining new release names. In short, beginning with VIC version 5.0.0, version numbers follow a pattern of 'MAJOR.MINOR.PATCH' where MAJOR versions represent when changes are included that are not backward compatible, MINOR versions represent when features and new functionality are added in a backwards-compatible manner, and PATCH versions represent when backwards-compatible

bug fixes are made.

As part of the development of VIC-5, we have implemented an updated, comprehensive documentation website for VIC (http://vic.readthedocs.io). While previous versions of the VIC model did have a documentation website, edits were only possible by the VIC administrator, and the documentation was maintained separately from the model source code. In VIC-5, the documentation content is part of the VIC source code repository and is hosted on ReadTheDocs, a public documentation

hosting service. A key benefit of the updated documentation is the ability to provide versioned documentation that matches specific model releases.

## 3 VIC-5 Simulations

The science, release, and performance tests described above were performed prior to the release of VIC-5. The tests were used to evaluate the final model version and were archived for future comparisons during the model development process. This

section, along with Table 2, describes specifics of the test simulations.

### 3.1 Science Tests

The scientific verification of the VIC-5 release included point comparisons of VIC simulations with 66 Ameriflux sites (Baldocchi et al., 1996, 2001; Bohn and Vivoni, 2016) and 448 SNOTEL sites. These sites cover a range of hydroclimates across North America, providing a sufficiently large collection of environments for testing purposes. For the VIC-5 release, these

datasets were used to ensure that the refactor did not significantly modify the model behavior relative to VIC-4. Future model development will use these archived test outcomes to motivate and evaluate scientific changes to the physical core of the model.

### 3.2 Release Tests

We have designed a series of release tests that may be used to benchmark the VIC model during the development and release process. The release tests are continental scale domain simulations performed using typical model settings. The specifics of the

release test simulations for the VIC-5 release are described in Table 2.

### 3.3 Performance Tests

We tested the parallel scaling performance of the *Image Driver* using a series of model simulations on the 50-km near equal-area pan-Arctic domain used as the standard configuration in RASM. This model domain is shown in Fig. 4 and is detailed in Table 2. Here we demonstrate the scaling performance of the parallelization scheme applied in VIC-5 where model throughput,

defined as simulated model years divided by wall-time in units of days, is shown as a function of the number of processors



used. Two RASM image driver model configurations were tested on the Thunder supercomputer at the U.S. Air Force Research
Laboratory (AFRL) Department of Defense (DoD) Supercomputing Resource Center (DSRC). Table 3 provides details on the
hardware specifications of this machine. The different model configurations represent two ends of the complexity spectrum
available in the VIC model. Configuration A was run in water balance mode, in which the energy balance is not explicitly
closed, resulting in a computationally less expensive model. Conversely, configuration B represents a much more complex
model, in which the energy and water budgets are solved together using an iterative solver. Both configurations were run
over the same model domain with the same meteorological forcings, with only differences in model configuration settings as
described above. Both model configurations were also tested using a hybrid OpenMP-MPI configuration with a combination
of shared memory threading and parallel processes. Because memory requirements are modest for VIC, we do not include any
memory profiling of the VIC parallelization schemes.

## 4   Results and Discussion

The inclusion of a comprehensive science testing framework within the VIC-5 repository, as introduced in Sect. 2.7, facilitates
continuous and repeatable validation of the model throughout the development process. Fig. 2 shows a summary graphic from
the SNOTEL component of the science testing package. This figure demonstrates that VIC-4 and VIC-5 are producing basically
identical ($r^2 > 0.99$) results for snow water equivalent (SWE) and that the two model versions have similar performance statistics
when compared to observations. Fig. 3 demonstrates an example comparison of latent and sensible heat from the FLUXNET
component of the science testing package, showing a comparison between VIC-5 and flux tower observations for a subset of
the sites included in the dataset. These automated diagnostics are particularly important throughout the model development
and evaluation process to ensure that changes to the core physics of the model do not result in unintentional outcomes in other
parts of the code. Additionally, these tests help provide baseline diagnostics for future VIC releases.

Improving the scaling performance of VIC-5 by adding parallel processing functionality enables VIC-5 to be used for
more computationally expensive applications, such as high-resolution modeling studies and coupled earth system models (e.g.
RASM) that have shorter coupling intervals (e.g. 20 minutes) than the timesteps that have traditionally been used in VIC (hourly
to daily). Fig. 5 demonstrates the scaling performance for the two model configurations. For the simpler model configuration
(A), VIC run with MPI shows high throughput, but extending parallelization beyond four to eight nodes (144 to 288 cores)
provides little speedup. This behavior is indicative of a program that is I/O limited. For the more complex model configuration
(B), VIC has significantly lower throughput, but scales efficiently beyond 32 nodes (1,152 cores). In this case, VIC spends
much more time inside the physics code than in the I/O code of the model. For example, a comparison of the timing logs for
two simulations that each use 144 MPI tasks shows that configuration A spends 48% on I/O operations, while configuration
B spends only 28%. In practice, the scaling performance for most parallel VIC applications is predominantly a function of a
computer's interconnect and disk speeds.

We also evaluated the scaling performance of the Hybrid OpenMP-MPI parallelization in VIC and found that these simula-
tions do not perform as well as the pure MPI implementation. Fig. 5 also includes the Hybrid OpenMP-MPI test simulations.



One probable explanation for this is that the MPI libraries on the Thunder supercomputer are optimized such that the overhead of switching between MPI and OpenMP is larger than the performance improvements gained by shared memory threading. We expect that for other combinations of model configuration, machine, compiler, and MPI and OpenMP libraries, this hybrid parallelization scheme will perform better than pure MPI.

5    Additional performance enhancement in scaling of the VIC *Image* and *CESM drivers* will likely occur through the application of parallel I/O. Collective parallel reading and writing of NetCDF files, made possible through the NetCDF-4 and HDF5 libraries, is expected to dramatically reduce the amount of data passed via MPI processes. For computers with high-performance parallel disk systems, parallel IO should allow the model to scale well beyond the limits shown in Fig. 5. The development of this feature has been reserved for later releases in the VIC-5 development track.

## 5    Conclusions

The VIC model continues to serve a broad user community. The VIC Users email list-serve (vic_users@u.washington.edu) has more than 420 active members, and the analytics from the VIC website and documentation indicate that there are upwards of one-thousand individual VIC users. The original VIC paper (Liang et al., 1994) continues to be widely cited in the hydrologic modeling literature with nearly 2,300 citations as of January 2018. The VIC source code was moved to a public GitHub repository in 2013. Since then over 160 users and developers have checked out "Forks". Additionally, we are aware of multiple ongoing developing efforts by the VIC community to implement new features. The open-source framework that we have implemented will facilitate mechanisms for the integration of model improvements from the greater VIC community into future VIC releases.

Beginning in 2014, the VIC model has been licensed with the open-source GNU General Public License version 2. The motivation behind making the model source code completely open-source was to encourage participation by the model user and development community-at-large, and to increase scientific transparency in the model development and application process (Ince et al., 2012). The VIC source code now uses the Git version control system (Torvalds and Hamano, 2010) and is publicly available on GitHub (https://github.com/UW-Hydro/VIC). While the majority of the VIC-5 development was done by researchers at the University of Washington, a number of features additions and bug fixes were made by unsolicited contributors. For example, the implementation of the RVIC streamflow routing extension within VIC (Sect. 2.4) was a community contribution and provides a poignant illustration of the benefits of convenient open-source collaboration.

The development of VIC-5 focused on refactoring the legacy source code to support a range of modern modeling applications. While the release of VIC-5 does not modify existing or add new process representations, it does modernize the supporting infrastructure in the model while preserving legacy functionality. The incorporation of state of the art software engineering tools, such as GitHub and continuous integration, will facilitate the ongoing development and maintenance of the model by the broader VIC developer community.





## 6   Code Availability

Appendix B describes the locations and license information for the the VIC source code and documentation. The source code, input data, and VIC configurations used as demonstrations for this paper, are provided in this GitHub repository: https://github.com/jhamman/VIC5_paper.

## 5   Appendix A: Key VIC Model Development Milestones

Since the original release of the VIC model and the original publication Liang et al. (1994), continued model development has added a series of new features and additional process representation. Table 4 highlights the key model development milestones and provides citations to the relevant literature.

## Appendix B: VIC Model Source Code and Documentation

### B1   Archive

- *Name*: VIC.5.0.1
- *Persistent identifier*: http://doi.org/10.5281/zenodo.267178
- *License*: GNU General Public License, version 2 (GPL-2.0)
- *Publisher*: Zenodo
- *Version published*: 5.0.1
- *Date published*: February 1, 2017

### B2   Code Repository

- *Name*: GitHub
- *Identifier*: https://github.com/UW-Hydro/VIC
- *License*: GNU General Public License, version 2 (GPL-2.0)
- *Date published*: February 1, 2018

### B3   Versioned Documentation

- *Name*: ReadTheDocs
- *Identifier*: http://vic.readthedocs.io/en/vic.5.1.0





- *License*: GNU General Public License, version 2 (GPL-2.0)

- *Date published*: February 1, 2018

*Author contributions.* J. Hamman, B. Nijssen, and T. Bohn performed the majority the source code reconfiguration. B. Nijssen provided overall oversight. J. Hamman designed the experiments with help from Y. Mao and D. Gergel to carry them out. J. Hamman and B. Nijssen
5 prepared the manuscript with contributions from all co-authors.

*Acknowledgements.* This research was supported, in part, by United States Department of Energy (DOE) grants DE-FG02-07ER64460 and DE-SC0006856 to the University of Washington, and grant 1216037 of the National Science Foundation's Science, Engineering, and Education for Sustainability (SEES) Program. Supercomputing resources were provided through the United States Department of Defense (DOD) High Performance Computing Modernization Program (HPCMP) at the Army Engineer Research and Development Center (ERDC)
10 and the Air Force Research Laboratory (AFRL). We thank Anthony Craig for his feedback during the design and implementation of the Image and CESM drivers. We also thank Wietse Franssen and Iwan Supit for their contributions to the streamflow routing extension.



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





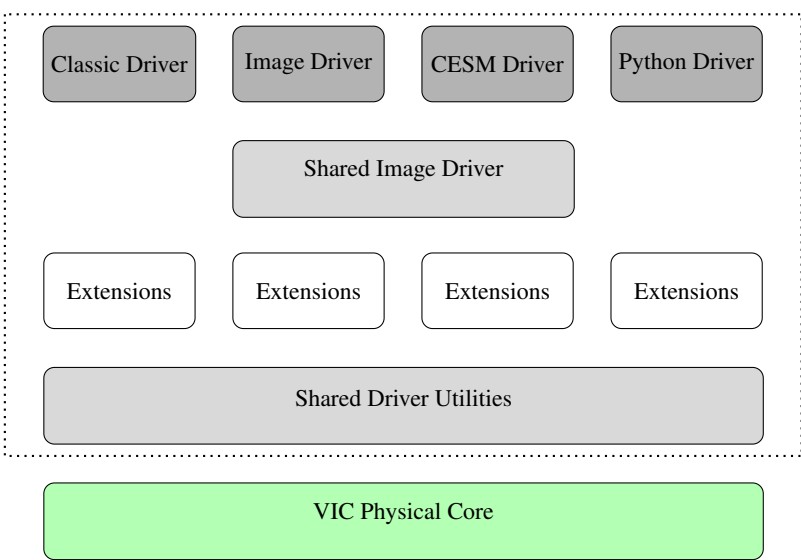

**Figure 1.** The reconfigured VIC-5 source code structure. The "Physical Core" includes the scientific modules of VIC (e.g. routines that simulate the fluxes and states of the model) and is used by each of the four drivers. The remainder of the VIC source code structure is considered driver-level code. Each driver uses components of the *Shared Driver* to minimize code duplication at the driver level. Likewise, the *Shared Image Driver* includes source code that is used by both the *Image* and *CESM drivers* (e.g. MPI and NetCDF utilities). Finally, the "Extensions" constitutes optional and driver specific sub-models.





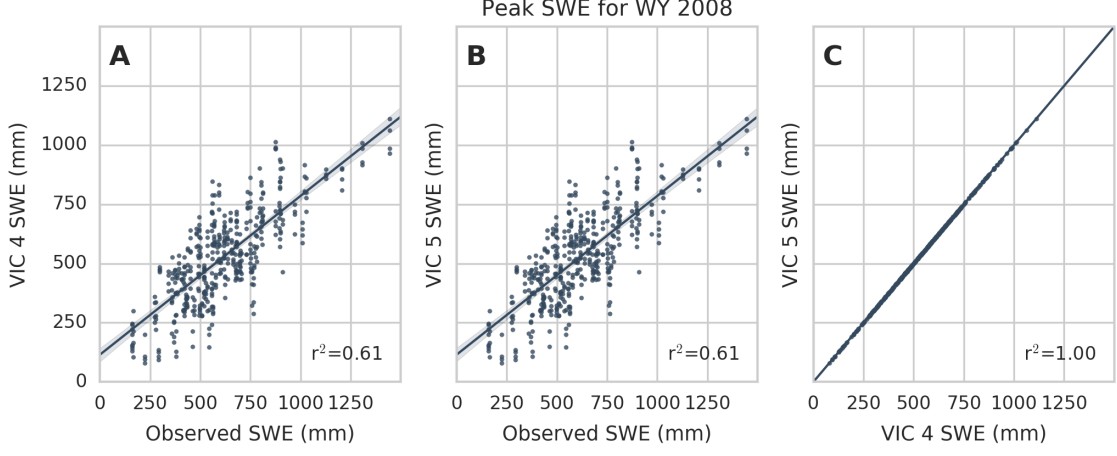

**Figure 2.** Example summary figure from the VIC-5 SNOTEL science tests. Panels A and B compare observed snow water equivalent (SWE) from SNOTEL sites to simulated SWE from VIC-4 (A) and VIC-5 (B). Panel C compares the results from the two VIC versions.



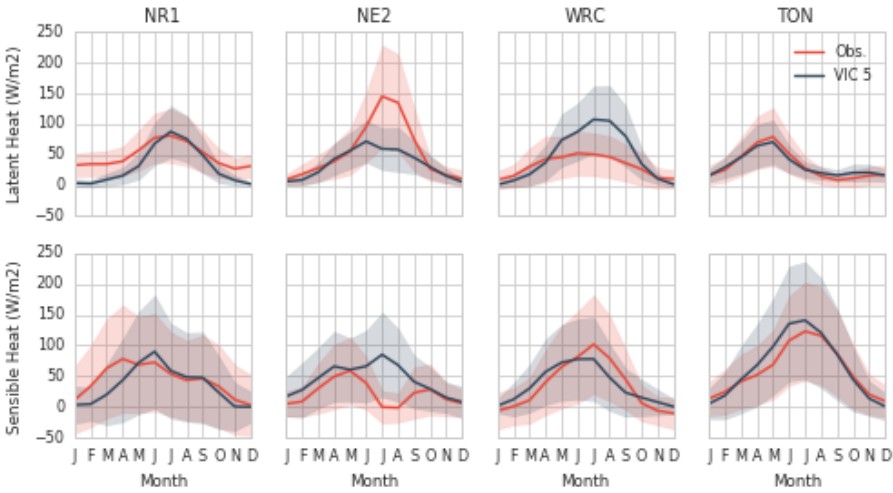

**Figure 3.** Example summary output from the VIC-5 science testing package comparing the annual cycle of latent heat (top) and sensible heat (bottom) for four sample locations for VIC-5 (blue) to FluxNet observations (red).

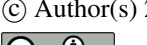

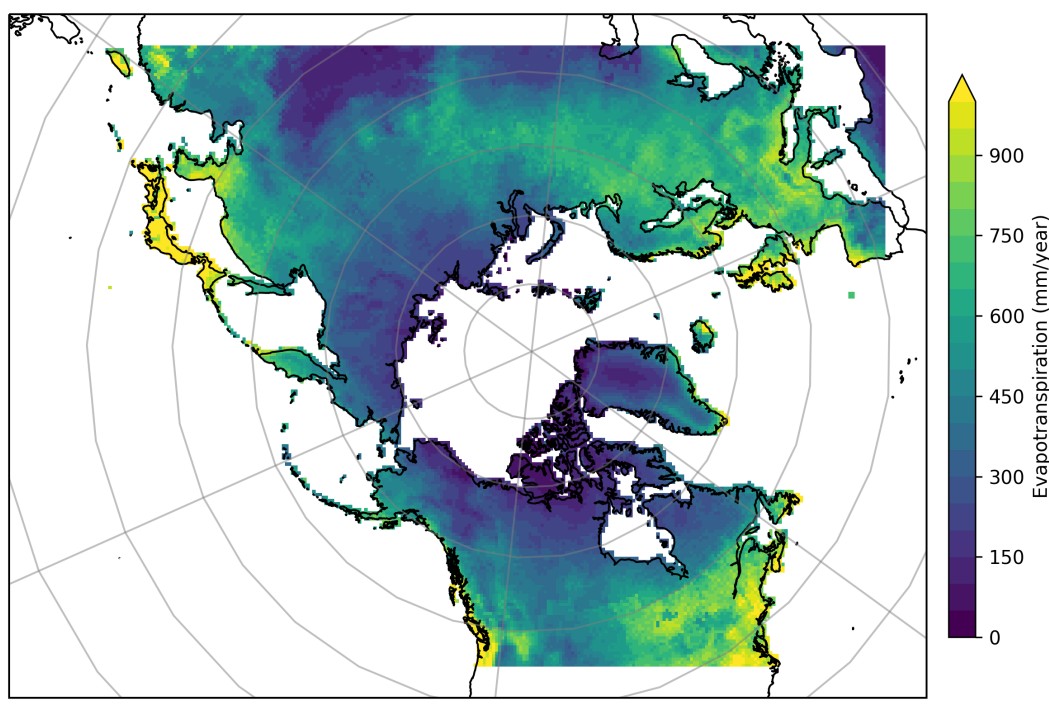

**Figure 4.** The 50-km near equal-area Regional Arctic System Model (RASM) domain. The model domain is comprised of 25,996 grid cells. Shading denotes mean annual evapotranspiration from a test simulation run with model configuration (B) as described in Section 3.





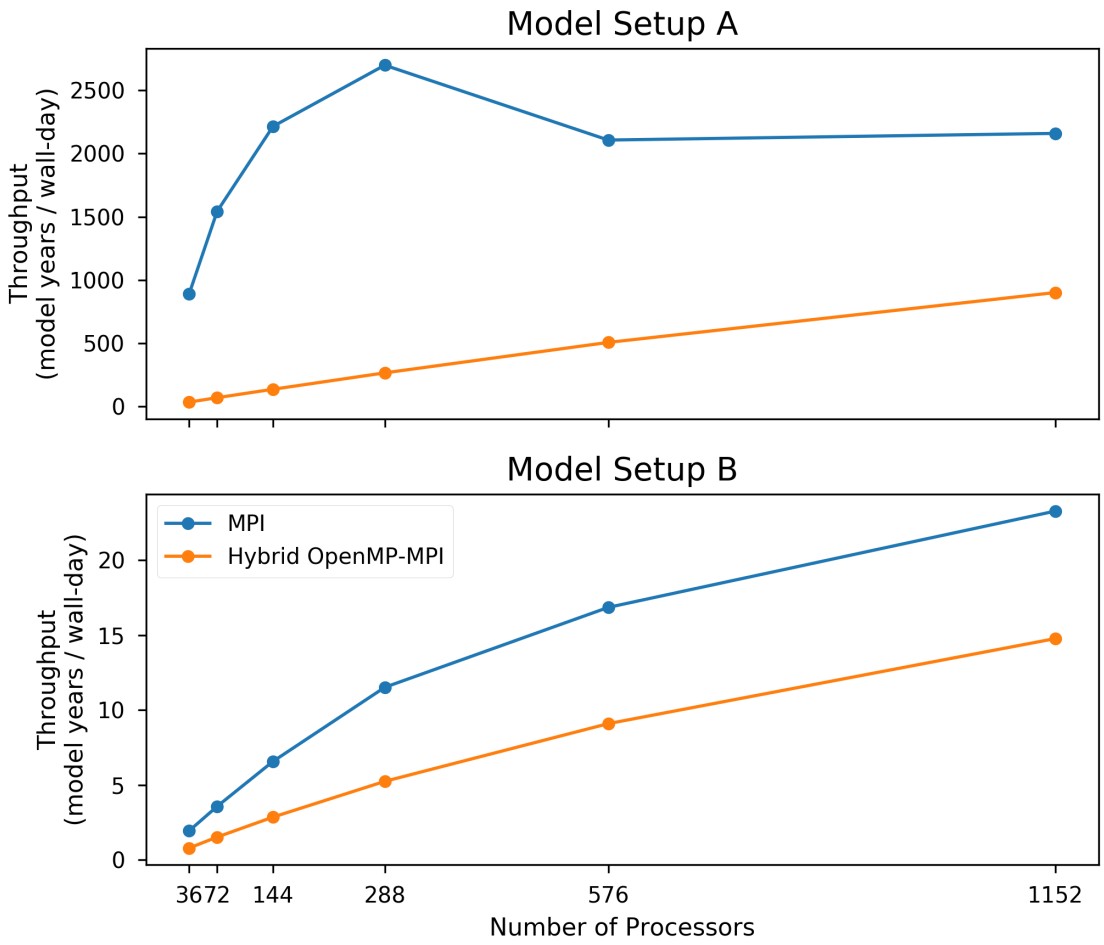

**Figure 5.** Model throughput for the two model configurations described in Section 3 run on the Thunder (SGI Ice X) supercomputer. The MPI only simulations are shown in blue and the Hybrid OpenMP-MPI simulations are shown in orange.



**Table 1.** Examples of VIC applications

| Data set construction | |
|---|---|
| US | Maurer et al. (2002) |
| Global | Nijssen et al. (2001a, c); Sheffield et al. (2006) |
| Arctic | Hamman et al. (2017) |
| **Historic trend analysis** | |
| Snow | Hamlet et al. (2005); Shi et al. (2011); Mote et al. (2005) |
| Land use/cover change | Matheussen et al. (2000) |
| Streamflow | Hamlet and Lettenmaier (2007); Hamlet et al. (2007); Tan et al. (2011) |
| Drought | Gao et al. (2011); Sheffield and Wood (2007, 2008); Sheffield et al. (2009); Wang et al. (2011); Nijssen et al. (2014) |
| **Data evaluation** | |
| Satellite precipitation | Nijssen and Lettenmaier (2004); Pan et al. (2010); Su et al. (2008) |
| Reanalysis | Maurer et al. (2001) |
| **Data assimilation** | |
| Snow | Andreadis and Lettenmaier (2006) |
| Soil moisture | Pan and Wood (2006) |
| **Forecasting and nowcasting** | |
| Droughts | Shukla et al. (2011) |
| Streamflow | Hamlet and Lettenmaier (1999); Li et al. (2009); Wood et al. (2002) |
| Predictability | Gebregiorgis and Hossain (2011); Maurer and Lettenmaier (2003) |
| **Climate change impact analysis** | |
| Hydrology | Barnett et al. (2005); Beyene et al. (2010); Nijssen et al. (2001b); Schewe et al. (2014) |
| Water resources | Christensen and Lettenmaier (2007); Das et al. (2011); Hamlet and Lettenmaier (1999) |
| **Coupled land-atmosphere modeling** | |
| US | Zhu et al. (2009) |
| Arctic | Hamman et al. (2016) |

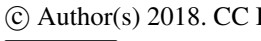



**Table 2.** Summary of tests and datasets.

| Test type | Driver | Domain | Grid cells | Resolution | Timestep | Forcings | Parameters | On Travis |
|---|---|---|---|---|---|---|---|---|
| **Build** | Classic, Image, CESM, Python | - | - | - | - | - | - | yes |
| **Unit** | Python | - | - | - | - | - | - | yes |
| **System** | Classic, Image, CESM | Stehekin | 16 | 1/8-deg. | 1-hour | Maurer et al., 2002 | Maurer et al., 2002 | yes |
| **Science** | Classic | SNOTEL | 448 point locations | point | 1-hour | in-situ observations | Maurer et al., 2002 | no |
| | | FLUXNET | 66 point locations | point | 1-hour | in-situ observations | Maurer et al., 2002 | |
| **Examples** | Classic, Image | Stehekin | 16 | 1/8-deg. | 1-hour | Maurer et al., 2002 | Maurer et al., 2002 | yes |
| **Release** | Image | Global | 61,345 | 1/2-deg. | 6-hour | CRU-NCEP | Adam et al., 2006 | no |
| | | N. America | 333,579 | 1/16-deg. | 3-hour | Livneh et al., 2015 | Livneh et al., 2015 | |
| | | RASM | 25,996 | 50-km | 3-hour | Sheffield et al., 2006 | Hamman et al., 2016 | |
| **Performance** | Image | RASM | 25,996 | 50-km | 3-hour | Sheffield et al., 2006 | Hamman et al., 2016 | no |



**Table 3.** Hardware used in VIC Image driver parallel scaling performance tests.

|  | **Thunder** |
|---|---|
| **Operated By** | Air Force Research Laboratory (AFRL) |
| **System** | SGI ICE X |
| **Peak PFlops** | 5.62 |
| **Parallel disk storage (Pbytes)** | 12.2 |
| **Total Nodes** | 3,216 |
| **Cores per Node** | 36 |
| **Core Type** | Intel Xeon E5-2699v3 |
| **Core Speed (GHz)** | 2.3 |
| **Memory/Node (Gbytes)** | 128 |
| **Memory Model** | Shared on node. Distributed across cluster. |
| **Interconnect Type** | FDR 14x InfiniBand / Enhanced LX Hypercube |



**Table 4.** Summary of major VIC developments present in VIC-5, focusing on improved process representation, and the model versions they were added in since the original publication of (Liang et al., 1994).

| **Vegetation** | |
|---|---|
| VIC.4.0.3 | Resistance factor approach to canopy resistance (Wigmosta et al., 1994) |
| VIC.4.2.0 | Fractional canopy coverage; daily timeseries of phenology (Bohn and Vivoni, 2016) |
| **Lakes and Wetlands** | |
| VIC.4.1.0 | Lake/wetland model (Cherkauer et al., 2003; Bowling and Lettenmaier, 2010) |
| **Soil** | |
| VIC.4.0.3 | Finite difference soil temperature scheme with frozen soil (Cherkauer and Lettenmaier, 1999) |
| VIC.4.0.3 | Exponential (quick-flux) soil temperature profile (Liang et al., 1999) |
| VIC.4.1.0 | Soil temperature statistical heterogeneity (Cherkauer et al., 2003) |
| VIC.4.1.0 | Implicit soil temperature scheme with optional exponential node distribution (Adam and Lettenmaier, 2008) |
| **Snow** | |
| VIC.4.0.3 | Elevation (snow) bands (Nijssen et al., 1997) |
| VIC.4.0.3 | Multi-layer snow pack (Cherkauer and Lettenmaier, 1999; Andreadis et al., 2009) |
| VIC.4.1.0 | Blowing snow sublimation (Bowling et al., 2004) |