# Peer review of "The Variable Infiltration Capacity Model, Version 5 (VIC-5): Infrastructure improvements for new applications and reproducibility"

_Geoscientific Model Development, 2018_

## Referee Comment (RC1) · T. Wagener (Referee) · 19 Apr 2018

The manuscript by Hamman et al. introduces a new version of the widely used and highly cited VIC model. It is, to my knowledge, the first time the model is properly published in a journal of the nature of GMD. This is a relatively short and well written manuscript, focusing on the latest version of VIC (5) and its implementation for distribution within the scientific community. The extensive use of VIC in the community already suggests that this will be a very welcome activity.

As such I have few comments:

[1] Is the African flood and drought monitor different from the global drought monitor? If yes, then maybe one could add this in the introduction section.

[2] The authors might have excluded it on purpose, but I would find it helpful to see the schematic figure of VIC that is widely used. This would be mainly helpful for people not familiar with VIC. My suggestion would be to include a version of it.

[3] Maybe the conclusions section could include a brief outlook paragraph in which the authors discuss what they see as the future evolution of VIC? It would be interesting to hear what the authors think is the future of this code as a scientific and/or operational tool for hydrology and water management.

[4] One issue that in the past has been problematic for LSMs is the detailed assessment of uncertainty (see Wood et al. vs Beven and Cloke discussion in WRR). I wonder what impact the re-structuring of the code in VIC-5 for the ability of modellers to undertake a detailed uncertainty analysis (especially of space-time fields)? Both in terms of memory requirements as well as in terms of model run times (on clusters).

[5] And finally, given the tremendous number of papers in which VIC is advanced and used, could the authors make some suggestions about what papers a new user should start with? Maybe a basic reading list. Where should a new user start after downloading VIC-5?

---

## Referee Comment (RC2) · Anonymous Referee #2 · 20 Apr 2018

This paper presents an overview of the recent changes to the widely-used VIC hydrology model, with a major reworking of its structure. The manuscript is well written and organized, and apart from a few minor issues I think it is in very good shape. The one aspect of the new model that was not described adequately was the extension system, and I would like to see a bit more detail on how the coupling with other models was done. Other minor issues that would need some clarification are outlined below:

* p. 2, l. 16: might want to clarify "coupled" to "two-way coupled". * p. 7, l. 10-11: have there been any tests and evaluation of the vectorization (if any is included) performance? This would be relevant to different types of processors such as the Intel

Xeon Phi ($\sim$60 cores). * p. 7, l. 30: would it have been possible to modify MT-CLIM instead in order to facilitate exact restarts? * p. 8, l. 12: are there any other pre-processing packages provided that can help with preparing e.g. soil information in the new format?

————————————————————

---

## Referee Comment (RC3) · Anonymous Referee #3 · 29 Apr 2018

General Comments:

This paper provides a comprehensive overview of the new updates and major revisions made to the Variable Infiltration Capacity (VIC) macroscale hydrology. This new version 5 of the model allows for different instances of the model driver, preserving legacy aspects of VIC and paving ways for coupling with other models, e.g., atmospheric models. A major change involves allowing the model to run in space first then in time, which was done in reverse in its legacy versions (prior to version 5). Also, the authors describe and provide some examples of several new test cases, which can be used for either unit and more scientific testing.

[Figure]

Overall it is very well written paper and provides many details on the way new model drivers and components can be run and interact in VIC. Not many papers prior to this have described the main software components of VIC, so in that of itself makes the paper more novel for the VIC and land surface model communities to reference in the future.

Specific Comments:

One area that is not addressed well in this paper is how the VIC distributed parameters (e.g., LAI or soils information) are handled and provided for the different drivers. The legacy "Classic" driver's original I/O is mentioned to still be supported, but how is that information changed (or not) for the other formats (e.g., NetCDF) to support the other space-before-time or "image-based" drivers. The VIC parameters for previous versions have been another problem for the community as they are hard to generate or customize for each user's needs. Please address how parameters can be generated (e.g., any tools) to support the "Image" type drivers and how some of the parameter tuning processes and practices (e.g., for soil parameters) can be used to optimize streamflow estimates. The handling of the parameters and preprocessing to other resolutions and grids is an important part of any model structure and release.

Minor Comments:

Page 2, lines 13-15: Part c): Previous VIC versions that have been run, prior to 5, were very slow, especially when run for large-scale simulations (e.g., NLDAS), since each gridcell was run separately through time (as referred to as "time-before-space" in the paper). Many of the other LSMs that you mention are not always run in "coupled" mode and tend to run faster than VIC in uncoupled mode. You may want to consider removing this statement or addressing the deficiency of it in relation to computation speed and how truly it was able to run faster.

Page 4, line 15: Please try to indicate what the acronym, "MT-CLIM", stands for.

Page 5, line 6: Does "RVIC" simply stand for "routing VIC"? Please include any additional information to specify this acronym.

Page 7, lines 30-32: The terminology convention of using "restarts" can be specific to certain communities, like the hydrometeorology community. It may be useful here to provide some short background on, first, what the "restart" represents and then to further specify what "exact" and "near-exact" indicate. Also, do all VIC versions, including 5, have an option to write out "instant" states and fluxes versus time-averaged?

Page 8, lines 9-13: The authors state here that the "input forcings must now have the same time step length as the model simulation" due to the removal of the MT-CLIM temporal disaggregation code, which has now become a separate preprocessing step. The reviewer wonders why temporal interpolation options have not been considered at the driver level for the forcing inputs, or why the MT-CLIM could not be rewritten to be incorporated more at the driver level. Please address in a statement or two.

Page 8, lines 25-27: Which libraries and compilers has the new VIC version been tested with? This information would be useful to the user-audience.

Page 8, lines 28-29: Are unit tests also available to be tested with the other drivers? What if changes to the drivers affect the unit tests? Please address this concern.

Page 9, lines 1-5: Why are the science-level tests only available with one driver type? Do you have science tests also set up for the other driver types?

Page 9, lines 1-15: Authors may want to mention that further descriptions of the different test figures and configuration setup are found also in Section 3.

Page 9, line 16: The "Travis CI" system is outlined for how it is used with VIC workflow and testing. Can the authors provide some additional information on the origins and source of Travis?

Page 11, lines 16-19, and Figure 3: Please either indicate here that the VIC-5 version performs at or above 0.99 r^2 values relative to previous VIC-4.x version(s), as you did

with the observed SNOTEL peak SWE comparison in Figure 2, or include a third time series of the VIC-4 version to show overlap with VIC-5.

Page 11, lines 24-29: Please add to this discussion how the information in Table 3 may relate to the tested parallelization performance metrics shown in Figure 5.

Figure 1: Please provide more information in the caption as to what is represented in the "Shared Driver Utilities". This part of the figure was not well addressed in the figure or accompanying documentation.

Figure 3: Mention in figure caption what the red and blue shading represent.

Table 4: Replace "they" with "that" and don't need the parentheses around "Liang et al. (1994)", just 1994.

---

## Author Comment (AC1) · 5 Jul 2018

**Author Response to Reviewer 1 (T. Wagener)**

*We thank the editor and reviewers for their comments, which helped improve this manuscript. Please note: responses to reviewer comments are italicized and bolded text below each reviewer's comments.*

The manuscript by Hamman et al. introduces a new version of the widely used and highly cited VIC model. It is, to my knowledge, the first time the model is properly published in a journal of the nature of GMD. This is a relatively short and well written manuscript, focusing on the latest version of VIC (5) and its implementation for distribution within the scientific community. The extensive use of VIC in the community already suggests that this will be a very welcome activity. As such I have few comments:

*Thank you for taking the time to review this paper. We have appreciated your comments.*

[1] Is the African flood and drought monitor different from the global drought monitor? If yes, then maybe one could add this in the introduction section.

*The Global drought monitor (Nijssen et al 2014) and the African drought monitor (Sheffield et al 2014) are, indeed, different systems. We have added a citation to the African drought monitor in table 1.*

[2] The authors might have excluded it on purpose, but I would find it helpful to see the schematic figure of VIC that is widely used. This would be mainly helpful for people not familiar with VIC. My suggestion would be to include a version of it.

*We appreciate the suggestion. Versions of the classic "VIC schematic" have been published many times before and we did not feel it was necessary to include in this paper. The VIC documentation website, which is linked to from this paper, does include the figure for interested readers.*

[3] Maybe the conclusions section could include a brief outlook paragraph in which the authors discuss what they see as the future evolution of VIC? It would be interesting to hear what the authors think is the future of this code as a scientific and/or operational tool for hydrology and water management.

*We have expanded our conclusions section to provide additional discussion on the future evolution of VIC. We have intentionally emphasized the role of the open-source community in future maintenance and development activities.*

[4] One issue that in the past has been problematic for LSMs is the detailed assessment of uncertainty (see Wood et al. vs Beven and Cloke discussion in WRR). I wonder what impact the re-structuring of the code in VIC-5 for the ability of modellers to undertake a detailed uncertainty analysis (especially of space-time fields)? Both in terms of memory requirements as well as in terms of model run times (on clusters).

*Our intent in this paper was to emphasize the new developments in and possible applications of VIC.5. Uncertainty analysis is one application that would be easier in VIC.5 when leveraging new features such as 1) configurable model parameters, 2) parallel computing enhancements that greatly speed up run and post-process times, and 3) improved I/O formats for both model parameters and model output. When combined, these new features allow a scientific user to more readily perform sensitivity analysis. We have highlighted this point in the manuscript.*

[5] And finally, given the tremendous number of papers in which VIC is advanced and used, could the authors make some suggestions about what papers a new user should start with? Maybe a basic reading list. Where should a new user start after downloading VIC-5?

*While the original Liang et al (1994) papers provide good context for the origins of the VIC model, significant model development has occurred since then and the early papers do not necessarily provide new users with a complete, or even accurate, description of the model. Even more so, overview papers describing VIC model development (such as this one) have been few and far between. We therefore suggest starting with the VIC documentation website. Users can easily read an overview of current model features and follow citations to specific papers describing the development of individual components of the mode according to their specific interest. This point has been emphasized more clearly in the manuscript.*

---

## Author Comment (AC2) · 5 Jul 2018

**Author Response to Reviewer 2**

*We thank the editor and reviewers for their comments, which helped improve this manuscript. Please note: responses to reviewer comments are italicized and bolded text below each reviewer's comments.*

This paper presents an overview of the recent changes to the widely-used VIC hydrology model, with a major reworking of its structure. The manuscript is well written and organized, and apart from a few minor issues I think it is in very good shape. The one aspect of the new model that was not described adequately was the extension system, and I would like to see a bit more detail on how the coupling with other models was done. Other minor issues that would need some clarification are outlined below:

*Thank you for taking the time to review this paper. We have appreciated your comments.*

*We have extended our description of both the extensions system and CESM coupling.*

\* p. 2, l. 16: might want to clarify "coupled" to "two-way coupled".

*We have changed the wording of this line to reflect that the previous versions of VIC didn't have to communicate with any other program (1 or 2 way coupling).*

\* p. 7, l. 10- 11: have there been any tests and evaluation of the vectorization (if any is included) performance? This would be relevant to different types of processors such as the Intel C1 Xeon Phi (~60 cores).

*None that we are aware of. We would be open to community contributions to the development and evaluation of vectorization performance metrics for VIC.5.*

\* p. 7, l. 30: would it have been possible to modify MT-CLIM instead in order to facilitate exact restarts?

*Technically, yes, it would have been possible to modify MT-CLIM instead of removing it. However, as we describe in the manuscript, the decision to remove MT-CLIM was made based on a number of factors, including model transparency, simplicity, and extensibility. One practical limitation with including MT-CLIM in VIC 5 was that its memory model would have incurred a significant memory footprint that was not practical in image-driver simulations.*

\* p. 8, l. 12: are there any other preprocessing packages provided that can help with preparing e.g. soil information in the new format?

*We are aware of a number of efforts across the VIC community to develop tools supporting the new VIC.5 file formats. We expect those to be published separate from this paper.*

---

## Author Comment (AC3) · 5 Jul 2018

**Author Response to Reviewer 3**

*We thank the editor and reviewers for their comments, which helped improve this manuscript. Please note: responses to reviewer comments are italicized and bolded text below each reviewer's comments.*

General Comments:

This paper provides a comprehensive overview of the new updates and major revisions made to the Variable Infiltration Capacity (VIC) macroscale hydrology. This new version 5 of the model allows for different instances of the model driver, preserving legacy aspects of VIC and paving ways for coupling with other models, e.g., atmospheric models. A major change involves allowing the model to run in space first then in time, which was done in reverse in its legacy versions (prior to version 5). Also, the authors describe and provide some examples of several new test cases, which can be used for either unit and more scientific testing.

Overall it is very well written paper and provides many details on the way new model drivers and components can be run and interact in VIC. Not many papers prior to this have described the main software components of VIC, so in that of itself makes the paper more novel for the VIC and land surface model communities to reference in the future.

*Thank you for taking the time to review this paper. We have appreciated your comments.*

Specific Comments:

One area that is not addressed well in this paper is how the VIC distributed parameters (e.g., LAI or soils information) are handled and provided for the different drivers. The legacy "Classic" driver's original I/O is mentioned to still be supported, but how is that information changed (or not) for the other formats (e.g., NetCDF) to support the other space-before-time or "image-based" drivers. The VIC parameters for previous versions have been another problem for the community as they are hard to generate or customize for each user's needs. Please address how parameters can be generated (e.g., any tools) to support the "Image" type drivers and how some of the parameter tuning processes and practices (e.g., for soil parameters) can be used to optimize streamflow estimates. The handling of the parameters and preprocessing to other resolutions and grids is an important part of any model structure and release.

*We agree that a new set of tools is needed in the "VIC ecosystem" to enable community wide adoption of new features in VIC, specifically the Image Driver. For example, the Image Driver now uses a netCDF file format for distributed input model parameters (e.g. soil and vegetation). We are aware of a number of efforts across the VIC community to develop tools supporting the new VIC.5 file formats. We expect those to be published separate from this paper.*

Minor Comments:

Page 2, lines 13-15: Part c): Previous VIC versions that have been run, prior to 5, were very slow, especially when run for large-scale simulations (e.g., NLDAS), since each gridcell was run separately through time (as referred to as "time-before-space" in the paper). Many of the other LSMs that you mention are not always run in "coupled" mode and tend to run faster than VIC in uncoupled mode. You may want to consider removing this statement or addressing the deficiency of it in relation to computation speed and how truly it was able to run faster.

*The statement in question is already conditioned on the comparison to larger coupled models. Moreover, our experience with previous versions of VIC taught us that the model was typically I/O bound – a fact that was exasperated by running each grid cell independent and writing separate files for each grid cell. Nevertheless, the computational resource required to run the model were in fact reasonably achieved without the requirement of large HPC systems.*

Page 4, line 15: Please try to indicate what the acronym, "MT-CLIM", stands for.

*We have indicated in the manuscript that MT-CLIM name is derived from the "Mountain Climate Simulator".*

Page 5, line 6: Does "RVIC" simply stand for "routing VIC"? Please include any additional information to specify this acronym.

*We have indicated in the manuscript that RVIC is short for "Routing VIC".*

Page 7, lines 30-32: The terminology convention of using "restarts" can be specific to certain communities, like the hydrometeorology community. It may be useful here to provide some short background on, first, what the "restart" represents and then to further specify what "exact" and "near-exact" indicate. Also, do all VIC versions, including 5, have an option to write out "instant" states and fluxes versus time-averaged?

*Thank you for this comment. We have clarified what we mean by the terms restarts/exact/near-exact/instant.*

Page 8, lines 9-13: The authors state here that the "input forcings must now have the same time step length as the model simulation" due to the removal of the MT-CLIM temporal disaggregation code, which has now become a separate preprocessing step. The reviewer wonders why temporal interpolation options have not been considered at the driver level for the forcing inputs, or why the MT-CLIM could not be rewritten to be incorporated more at the driver level. Please address in a statement or two.

*The specific reasons MT-CLIM was omitted VIC at the driver level are described in the numbered list in section 2.6.*

Page 8, lines 25-27: Which libraries and compilers has the new VIC version been tested with? This information would be useful to the user-audience.

*VIC is routinely tested with GNU (C and Fortran) and Clang (C) compilers, and the Open-MPI, and netCDF libraries. Though not part of the continuous integration system for licensing reasons, VIC has also been tested with Intel, PGI, and Cray compilers and the MPICH and Intel-MPI libraries. We have indicated these details to the manuscript.*

Page 8, lines 28-29: Are unit tests also available to be tested with the other drivers? What if changes to the drivers affect the unit tests? Please address this concern.

*At the time of writing, VIC's unit tests only test the physical core of the model, and the shared driver code. The system tests are designed to test the driver functions and they are tested using a similar python testing framework. This point has been clarified in the manuscript.*

Page 9, lines 1-5: Why are the science-level tests only available with one driver type? Do you have science tests also set up for the other driver types?

*We have scoped the science test category for those simulations that can be compared directly to observations (i.e. from flux towers). Within the system tests are configurations that test for equivalency between drivers (image vs. classic). By the transitive relation, we can say that the image driver will behave just like the classic driver. Therefore, running the science tests again for other drivers is not necessary. However, the release tests do mainly target the image driver and provide the opportunity for additional verification through model evaluation. We have tried to clarify these points in the manuscript.*

Page 9, lines 1-15: Authors may want to mention that further descriptions of the different test figures and configuration setup are found also in Section 3.

*Thank you for the suggestion. We have pointed to section 3 in this case.*

Page 9, line 16: The "Travis CI" system is outlined for how it is used with VIC workflow and testing. Can the authors provide some additional information on the origins and source of Travis?

*Travis CI is a hosted service for testing open-source software. The TravisCI Wikipedia page (https://en.wikipedia.org/wiki/Travis_CI) and the "Travis CI About Page" (https://about.travis-ci.com/) provide good overviews of the service, its typical uses, and its adoption across the software development community.*

Page 11, lines 16-19, and Figure 3: Please either indicate here that the VIC-5 version performs at or above 0.99 r^2 values relative to previous VIC-4.x version(s), as you did C3 with the observed SNOTEL peak SWE comparison in Figure 2, or include a third time series of the VIC-4 version to show overlap with VIC-5.

*We have added the VIC.4 results to this figure. The reviewers may note that the r2 values comparing VIC-4 and VIC-5 are 1.00 for all sites.*

Page 11, lines 24-29: Please add to this discussion how the information in Table 3 may relate to the tested parallelization performance metrics shown in Figure 5.

*We have elaborated on our previous point that as VIC is scaled out, it becomes more and more I/O bound. Therefore, a more performant disk system (e.g. fast SSDs) along with faster interconnects (e.g. infiniband) would likely yield improved scalability.*

Figure 1: Please provide more information in the caption as to what is represented in the "Shared Driver Utilities". This part of the figure was not well addressed in the figure or accompanying documentation.

*We have added a brief paragraph in section 2.3 describing what we mean by the Shared Driver Utilities.*

 Figure 3: Mention in figure caption what the red and blue shading represent.

*We have added a sentence describing the meaning of this shading (interannual variability represented as +/- 0.5 standard deviation).*

Table 4: Replace "they" with "that" and don't need the parentheses around "Liang et al. (1994)", just 1994.

*We have correct these two typos.*